# Association between *Angiotensin-converting enzyme (ACE) insertion/deletion* polymorphism and hypertension in a Ghanaian population

**Precious Bonney** [1]*, **Christian Obirikorang**[2], **Lawrence Quaye**[1], **Peter Paul Dapare**[1], **Yussif Adams**[1], **Grace Bourawono**[1], **Mercy Akaluti**[1], **Jemimah Duodu**[1]

**1** Department of Biomedical Sciences, University for Development Studies, Tamale, Ghana, **2** Department of Molecular Medicine, Kwame Nkrumah University of Science and Technology, Kumasi, Ghana

* preciousbonney903@yahoo.com

**Data Availability Statement:** All relevant data are within the manuscript and its Supporting Information files.

## Abstract

### Background

Genetic modifications in the renin-angiotensin aldosterone system (RAAS) have been suggested to play a key role in the pathophysiology of hypertension. The insertion/deletion polymorphism of angiotensin-converting enzyme (ACE) gene phenomenon and its relationship with essential hypertension has not been explored within the Ghanaian population. This study aims to determine the relationship between the ACE I/D polymorphism and the risk of essential hypertension among patients seeking medical attention.

### Methods

This case-control study was conducted at the Tamale Central Hospital in Ghana. A total of 144 study participants comprising 72 hypertensive patients and 72 normotensive individuals were recruited from May to July 2022. The modified WHO step questionnaire for chronic diseases was used to collect ACE concentrations and electrolytes were estimated and molecular testing conducted using to identify genotypes. To compare continuous variables between two groups and among multiple groups, the Student's t-test and analysis of variance (ANOVA) were used respectively. Genotype and allele frequencies were determined through direct counts, while differences in the distribution of alleles and genotypes between groups were estimated using chi-squared test.

### Results

The distribution of DD genotype and D allele respectively was 26.4% and 54% in hypertensives and 50% and 72% in normotensives. DD genotype significantly increased the risk of hypertension after adjusting for age and BMI (aOR = 8.52, 95% C.I = 1.22–59.6). In the recessive model, the risk of hypertension increased four times in subjects with the DD genotype (aOR = 4.09, 95% CI = 1.28–13.05). ACE levels were significantly elevated among hypertensives compared to controls, but did not significantly differ between the DD genotype and II+ID genotypes among hypertensives and normotensive subjects.

**Funding:** The author(s) received no specific funding for this work.

**Competing interests:** The authors have declared that no competing interests exist.

## Conclusion

The study shows that the presence of the DD genotype is strongly associated with an increased risk of hypertension in the Ghanaian population.

## 1. Introduction

Hypertension is defined by consistent elevated blood pressure above 140mmHg and 90mmHg for systolic and diastolic blood pressures respectively [1]. Hypertension is a prevalent and growing health problem worldwide, with an estimated global prevalence of 40% among adults 18 years and older, with majority of the prevalence burden in Africa [2], hypertension is a notable leading cause of death. The prevalence of hypertension is expected to increase by 80% in developing countries by 2025 [3]. In Ghana, the incidence of hypertension is increasing among varied populations [4, 5] and is one of the main causes of hospitalisation and death in Ghana [6].

Numerous candidate genes have been implicated in the development of hypertension, with genes in the renin-angiotensin aldosterone system (RAAS) being the focal point of investigation [7–9]. The RAAS is a critical regulator of sodium and water balance and consequently blood pressure. The angiotensin-converting enzyme (ACE) gene is a 21kb sequence located on chromosome 17q23. It encodes the enzyme which catalyses the conversion of angiotensin I to angiotensin II [7, 10]. Angiotensin II is the effector peptide of RAAS and causes vasoconstriction, sodium and water reabsorption, and elevation of aldosterone levels [7, 11]. The ACE gene comprises 26 exons and has many polymorphisms, notably an insertion or deletion (I/D) of a 276 bp Alu repetitive sequence in intron 16 which has been reported to influence levels of ACE [12, 13], thus leading to the postulation that ACE I/D is a possible cause in the development of hypertension.

The insertion/deletion polymorphism has been widely investigated because of its potential influence on the interpersonal variation of ACE levels however these studies have yielded conflicting results. Numerous studies have reported a significant risk in the development of hypertension associated with the ACE D allele and DD genotype among Nigerians [14], Indians [12], Bangladeshi [15] and Chinese populations [9, 10]. In contrast, studies among Tunisians [16, 17], Spanish [18]), Chinese [7] and Bangladeshi [19] did not show this association with another study that revealed an inverse association [20].

Due to significant contradictory findings in the literature among different populations, further studies should be conducted to determine the association between the ACE I/D polymorphism and susceptibility to hypertension conducted. However, there is scarcity of published data among the Ghanaian population. Therefore, this study aimed to determine the association between the ACE I/D polymorphism and hypertension among patients attending Tamale Central Hospital, Tamale Ghana.

## 2. Materials and methods

### 2.1 Study design

This case-control study was conducted at Tamale Central Hospital, a regional hospital in Northern region of Ghana. This hospital includes a hypertensive clinic manned by a specialist for the treatment and follow-up of hypertensives.

## 2.2 Study participants

All essential hypertensive patients were recruited as cases and apparently healthy individuals as normotensive controls between 1st May 2022 and 31st July 2022. The source population comprised people visiting the hospital. Cases included individuals on follow-up who had been on antihypertensive medication, while controls comprised apparently healthy normotensive blood donors. The ages of study participants ranged between ages of 18 and 60 years. Individuals who had clinically confirmed comorbidities such as tuberculosis, diabetes mellitus, liver disease, pregnancy-induced hypertension, renal disease, inflammatory disease, thyroid disease, or secondary hypertension were excluded.

## 2.3 Ethical considerations

Ethical clearance was sought from and a written approval was given by the Institutional Review Board of the University for Development Studies with reference number UDS/RB/0009/22. Participation in the study was voluntary, the rights of participants as study participants were explained to participants before they were recruited only after giving a verbal consent form and signing or thumbprinting on a written informed consent form. Rights as research subjects, anonymization and confidentiality of data was clearly explained to the study participants to enable them to take a decision to participate in the study.

## 2.4 Data collection procedure

The study employed semi-structured interviewer-administered questionnaires (WHO Step questionnaire for chronic diseases) and medical records review to collect socio-demographic data. Weight (in kg) and height (in meters) were measured using a standard balance and stadiometer which were used in calculating the body mass index (BMI). BMI was classified into four categories according to the WHO (2008) guidelines: underweight ($<$18.5 kg/m$^{-2}$), normal (18.5–24.9 kg/m$^{-2}$), overweight (25.0–29.9 kg/m$^{-2}$) and obese ($\geq$30.0 kg/m$^{-2}$).

Blood pressure readings were taken using a sphygmomanometer on the midpoint of the left arm in a seated position with arm support. Participants were made to rest for a minimum of 5 to 10 minutes before blood pressure measurements were taken. Two readings were recorded at a minimum interval of 5 minutes, and the average value was considered as the true value.

## 2.5 Blood samples collection and genetic analysis

Five (5) ml of venous blood was drawn from each study participant under aseptic conditions and aliquoted equally into vacutainer gel separator and tripotassium ethylenediaminetetraacetic acid (K$_3$EDTA) tubes respectively. Blood in the K$_3$EDTA tube was stored at -20˚C for molecular studies. Serum in the gel separator tube was spun at 1000rpm for 15 minutes and further aliquoted into two cryotubes for lipid profile and the remaining aliquote stored for angiotensin-converting enzyme estimation through enzyme-linked immunosorbent assay (ELISA).

**2.5.1 Biochemical analysis.** Biochemical assays (HDL-cholesterol and electrolytes) were estimated using the Mindray BS-240 autoanalyzer (Thermo Scientific, Finland). Serum ACE levels was estimated using a solid phase sandwich ELISA commercial kit (Monobind Inc USA.). All ELISA quantifications were performed according to manufacturer's protocol, read on a microplate reader (PoweAn WHYM200) at a wavelength of 450 nm, and reported in ng L$^{-1}$.

**2.5.2 Genomic DNA extraction and isolation.** DNA extraction was performed using the non-enzymatic salt extraction method described by [21], taking 300 μL into

**Table 1. Primer sequence used in PCR amplification of ACE I/D polymorphism.**

| Polymorphism | Primer sequence | Genotype | Amplicon size(bp) |
|---|---|---|---|
| *I/D* | | | |
| Forward | 5'-CTGGAAGAGACCACTCCCATCCTTTCT'-3 | DD | 190 |
| Reverse | 5'GATGTGGCCATCACATTCGTCAGAT3' | II | 490 |
| | | ID | 490 + 190 |

ethylenediaminetetra-acetic acid (EDTA) blood and transferring it to a 1.5 ml sterilised eppendorf tube. Red blood cell lysis buffer (RBC) solution was used to lyse and eliminate RBCs. Likewise, white blood cells were lysed using a solution of a nucleic lysis buffer. Then, a high-concentration salt (6M NaCl) was added to precipitate and remove proteins. DNA was precipitated by cooling using isopropanol and was washed with 70% cold ice ethanol. The genomic DNA was then dissolved with a Tris-EDTA buffer (TE) and stored at -21°C until use. The quality of isolated genomic DNA was confirmed by measurement of its absorption ratio at 260/280nm.

**2.5.3 Polymerase chain reaction.** Participant genotype was determined using polymerase chain reaction (PCR) with specific PCR primers flanking the polymorphic region outside the Alu sequence in intron 16 (**Table 1**). The final volume (25 μl) of the PCR reaction mixture was prepared with 10 pmol of forward and reverse primers (**Table 1**), 1.5 mM of $MgCl_2$, 0.2 mM of each dNTP, 1.0 units of Taq polymerase, 2 μL of template DNA and water. The DNA was amplified for 30 cycles The cycles were denaturation at 94°C for 30 seconds, annealing at 58°C for 30 seconds, extension at 72°C for 45 seconds, and final extension at 72°C for 9 minutes. The PCR product was held at 4°C until it was analysed by agarose gel electrophoresis. The ACE (I/D) PCR amplicons were electrophoretically separated for 45 minutes on 2% agarose gel. Band sizes observed as shown in **Fig 1** with an ultraviolet (UV) transilluminator were interpreted according to **Table 1** below.

## 2.6 Data analysis

Data was analyzed with Statistical Package for Social Sciences (SPSS v26.0) and Graphpad Prism v8.0. Continuous variables were reported as means and standard deviations (SD). To compare continuous variables between two groups and among multiple groups, the Student's t-test and analysis of variance (ANOVA) were used respectively. Genotype and allele frequencies were determined through direct counts, while differences in the distribution of alleles and genotypes between groups, as well as deviations from Hardy-Weinberg equilibrium, were evaluated using the chi-square test. The relationship between genetic variations and hypertension was studied using logistic regression, which estimates the odds ratio (OR) and the 95% confidence interval (95% CI). All tests were considered statistically significant at $p < 0.05$.

## 3. Results

A total of 144 participants were recruited, comprising 72 normotensives and 72 hypertensives. Hypertensives were significantly older (57 yrs) than normotensives (30 yrs). Majority of study participants were female (65.3%), had no formal education (40.3%), and were married (57.3%). Most hypertensives did not consume alcohol (94.4%) or smoke tobacco (93.1%) (**Table 2**). Most hypertensives did not exercise regularly (73.6%) and had significantly higher body mass index (26.5±4.4 kg/m$^{-2}$) compared to the normotensive group (23.3±3.8kg/m$^{-2}$). Hypertensives had significantly lower HDL-cholesterol and sodium concentrations (**Table 3**).

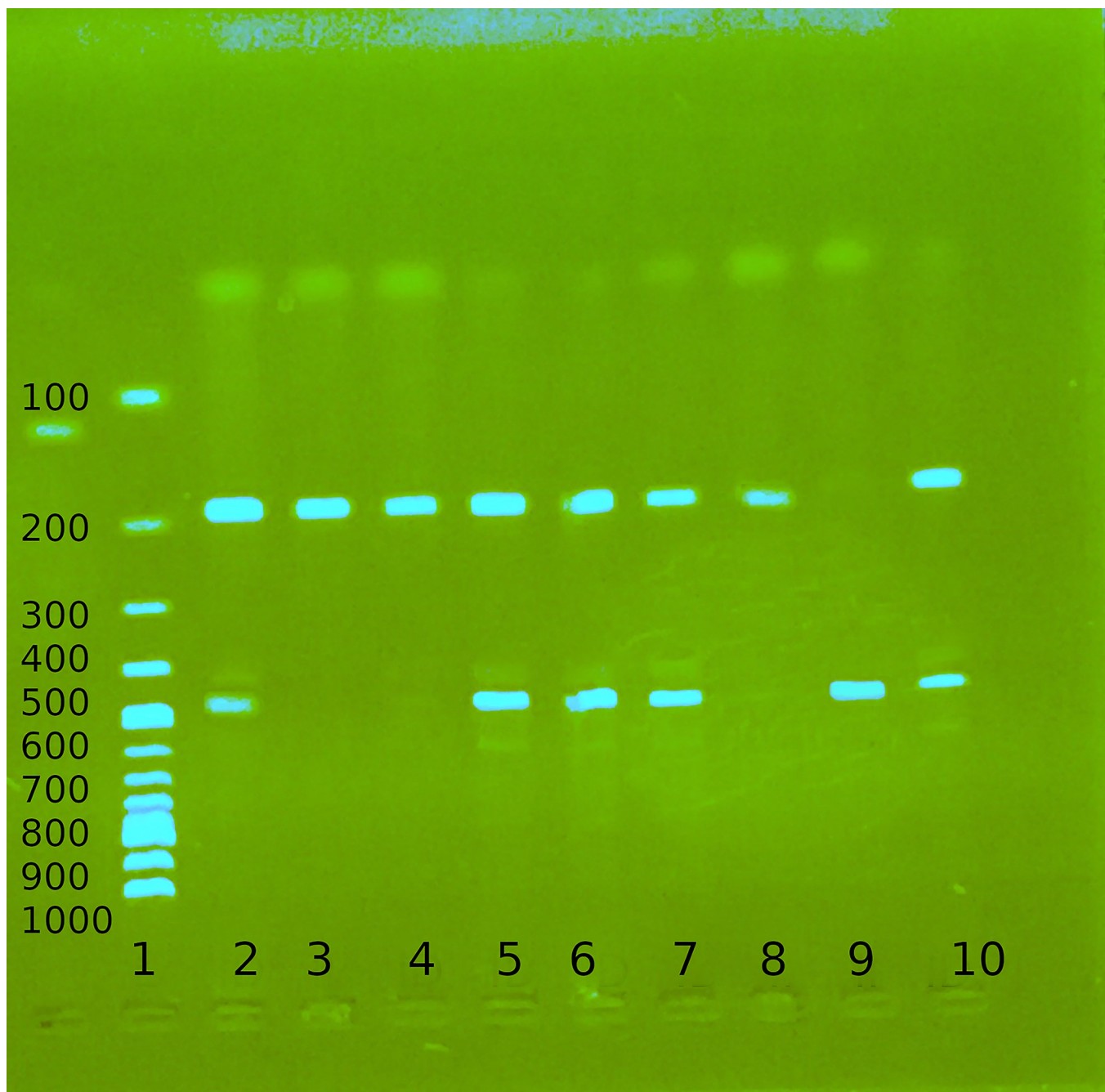

**Fig 1. Gel electrophoresis sample for ACE insertion/deletion polymorphism.** Lane 1-100bp ladder, Lane 2,5,6,7 and 10- heterozygous ID genotype, Lane 3, 4 and 8-homozygous deletion (DD) genotype, Lane 9-homozygous insertion (II) genotype.

Table 4 compares the proportions of genotypes between hypertensives and normotensives. The proportions of II and ID genotypes were significantly higher in hypertensives than normotensives, but the DD genotype was higher among normotensives.

Table 5 shows the results for the test for Hardy-Weiberg equilibrium. No significant differences were recorded in both hypertensives and normotensives and showing normal distribution of alleles and genotypes in this study.

**Table 2. Sociodemographic characteristics of participants.**

| Variable | Total N(%) | NTN N = 72 | HTN N = 72 | Chi-squared | P-value |
|---|---|---|---|---|---|
| **Sex** | | | | | |
| Male | 50(34.7) | 37(51.4) | 13(18.1) | 17.65 | <0.0001 |
| Female | 94(65.3) | 35(48.6) | 59(81.9) | | |
| **Educational status** | | | | | |
| No education | 58(40.3) | 6(8.3) | 52(72.2) | | |
| Primary | 8(5.6) | 3(4.2) | 5(6.9) | 68.6 | <0.0001 |
| High school | 26(18.1) | 18(25.0) | 8(11.1) | | |
| Tertiary | 52(36.1) | 45(62.5) | 7(9.7) | | |
| **Marital status** | | | | | |
| Single | 48(33.6) | 47(66.2) | 1(1.4) | 71.7 | <0.0001 |
| Married | 82(57.3) | 24(33.8) | 58(80.6) | | |
| Divorced | 2(1.4) | 0(.00) | 2(1.4) | | |
| Widow | 11(7.7) | 0(0.00) | 11(7.7) | | |
| **Alcohol consumption** | | | | | |
| Yes | 14(9.7) | 10(13.9) | 4(5.6) | 2.85 | 0.158 |
| No | 130(90.3) | 62(86.1) | 68(94.4) | | |
| **Tobacco smoker** | | | | | |
| Yes | 9(6.3) | 4(5.6) | 5(6.9) | 0.119 | 0.500 |
| No | 135(93.8) | 68(94.4) | 67(93.1) | | |
| **Exercise** | | | | | |
| Yes | 41(28.5) | 22(30.6) | 19(26.4) | 0.307 | 0.356 |
| No | 103(71.5) | 50(69.4) | 53(73.6) | | |
| **Family history of HTN** | | | | | |
| Yes | 63(44.1) | 31(43.7) | 32(44.4) | 0.009 | 0.535 |
| No | 80(55.9) | 40(56.3) | 40(55.6) | | |
| **BMI** | | | | | |
| Normal | 70(48.6) | 42(58.3) | 28(38.9) | | |
| Underweight | 6(4.2) | 4(5.6) | 2(2.8) | 14.44 | 0.002 |
| Overweight | 50(34.7) | 24(33.3) | 26(36.1) | | |
| Obese | 18(12.5) | 2(2.8) | 16(22.2) | | |

Data expressed as counts and proportions. Categorical data was compared using Chi-squared ($\chi^2$).

**Table 3. Biochemical characteristics of study participants.**

| Variable | NTN | HTN | P-value |
|---|---|---|---|
| | N = 72 | N = 72 | |
| Age (yrs) | 29.8±12.7 | 56.8±11.9 | <0.0001 |
| BMI (Kg/m$^2$) | 23.3±3.8 | 26.5±4.4 | <0.0001 |
| SBP (mmHg) | 117.2±18.4 | 146.9±21.3 | <0.0001 |
| DBP (mmHg) | 75.5±13.0 | 90.3±12.1 | <0.0001 |
| Heart rate (bpm) | 73.9±18.6 | 82.1±15.9 | <0.0001 |
| HDL-Chol (mmol/L) | 1.9±0.9 | 1.65±0.63 | 0.056 |
| Sodium (mmol/L) | 148.9±9.0 | 136.1±9.6 | <0.0001 |

Data presented as mean and standard deviation (mean ± SD), p-value computed using independent t-test.

**Table 4. Frequency distribution of genotypes and ACE alleles (I/D) between hypertensive and controls.**

| Genotype | NTN | HTN | chi-square ($\chi^2$) | p-value |
|---|---|---|---|---|
| II | 4(5.6) | 13(18.0) | 10.91 | **0.004** |
| ID | 32(44.4) | 40(55.6) | | |
| DD | 36(50.0) | 19(26.4) | | |
| **Dominant model** | | | | |
| ID+DD | 68(94.4) | 59(81.9) | 5.4 | **0.020** |
| II | 4(5.6) | 13(18.1) | | |
| **Recessive model** | | | | |
| DD | 36(50.0) | 19(26.4) | 8.5 | **0.003** |
| II+ID | 36(50.0) | 53(26.4) | | |
| **Co-dominant model** | | | | |
| ID | 32(44.4) | 40(55.6) | 1.78 | 0.182 |
| II+DD | 40(55.6) | 32(44.4) | | |
| **Allele** | | | | |
| I | 40(27.8) | 66(45.8) | 9.33 | **0.002** |
| D | 104(72.2) | 78(54.2) | | |

Data presented as absolute counts and proportions. P-value computed using the Chi-squared test.

Table 6 shows the logistic regression analysis to assess the risk associated with ACE genotypes. Significant associations between dominant, recessive and allelic models and hypertension. The DD genotype reduced the risk of developing hypertension, however after adjustment for age and BMI, the DD genotype significantly increased the risk of hypertension by 8-fold (OR = 8.52, 95% CI = 1.22–59.6) after. Similarly, the recessive model also increased the risk of developing hypertension four times after adjustment (OR = 4.09, 95% CI = 1.28–13.05).

Fig 2 shows the comparison of ACE levels between study groups(normotensives and hypertensives) and between genotypes. Elevated median ACE concentration was observed among hypertensives compared to normotensives. ACE concentrations were compared between DD genotype and (II+ID) genotypes of the hypertensive participants which showed no significant

**Table 5. Test for Hardy-Weinberg equilibrium among study participants, controls and hypertensives.**

| Genotype | Observed freq. | Expected freq. | Chi-squared($\chi^2$) | P-value | Allele | Frequency |
|---|---|---|---|---|---|---|
| **Total** | | | | | | |
| II | 17 | 19.7 | | | I | 0.37 |
| ID | 72 | 67.1 | 0.812 | 0.666 | D | 0.63 |
| DD | 55 | 57.2 | | | | |
| **Total** | **144** | **144** | | | | |
| **Controls** | | | | | | |
| II | 4 | 5.6 | | | I | 0.28 |
| ID | 32 | 29.1 | 0.791 | 0.673 | D | 0.72 |
| DD | 36 | 37.3 | | | | |
| **Total** | **72** | **72** | | | | |
| **Hypertensives** | | | | | | |
| II | 13 | 15.0 | | | I | 0.46 |
| ID | 40 | 36.0 | 0.902 | 0.637 | D | 0.54 |
| DD | 19 | 21.0 | | | | |
| **Total** | **72** | **72** | | | | |

**Table 6. Logistic regression analysis of the ACE polymorphism.**

| Genotype | cOR | p-value | 95%CI | adj OR | p-value | 95% C.I |
|---|---|---|---|---|---|---|
| II | Ref | | | Ref | | |
| ID | 0.385 | 0.123 | 0.114–1.294 | 2.48 | 0.382 | 0.38–16.1 |
| DD | **0.162** | 0.004 | 0.046–0.567 | **8.52** | **0.031** | **1.22–59.6** |
| **Dominant** | | | | | | 0.71–24.99 |
| (DD+ID) vs II | 0.267 | 0.027 | 0.083–0.863 | 4.2 | 0.114 | |
| **Recessive** | | | | | | **1.28–13.05** |
| DD vs (ID+II) | 0.358 | 0.004 | 0.178–0.721 | **4.09** | 0.017 | |
| **Co-dominant** | | | | | | 0.168–1.45 |
| ID vs (II+DD) | 1.56 | 0.183 | 0.81–3.02 | 0.494 | 0.200 | |

cOR: Crude odds ratio, AOR: adjusted odds ratio for age and BMI

difference. Similarly, no significant differences in ACE concentrations was observed between the DD genotype and (II+ID) genotypes among normotensive participants.

## 4. Discussion

Hypertension is one of the most common health problems in the world and causes high cardiovascular morbidity and mortality [9]. The influence of ACE polymorphisms on genetic hypertension is not fully understood. Thus, the present study aims to explore the distribution and association of ACE polymorphisms with hypertension among Ghanaian patients and corresponding healthy controls were explored in this study.

The Hardy-Weinberg test for equilibrium (HWE) for both control and patients' groups were satisfied. P-values < 0.05 for the HWE model reveals abnormal distribution of genotypes within a population of interest as a result of genetic events such as inbreeding, copy number variation and purifying selection [22].

The present study found a significant association between the ACE (I/D) polymorphism and hypertension. The DD genotype increased the risk of hypertension by 8 times compared to participants with the II genotype after adjusting for age and BMI. The significant increase in susceptibility to hypertension by subjects with DD genotype confirms the reports among Nigerians [14], Chinese [9], Burkinabe [23], South Indians [12] and Ethiopians [24]. Furthermore, a meta-analysis conducted by [25] also reported a significant pooled increase susceptibility to hypertension among Africans with the DD genotype. Studies conducted in large populations such as the Indian [26], Sikh [27] and African-American [28] populations also found the pivotal role played by the ACE I/D polymorphism which contrast findings of [29] who found no association between polymorphisms of the ACE gene and hypertension. The significant increase in the risk of developing hypertension is attributable to the deletion of the 287-bp sequence in intron 16 which plays a role in the regulation of transcriptional activity. The deletion results in an increase in the transcriptional activity and thus increases ACE gene expression [30, 31] resulting in higher ACE levels among hypertensives as observed in the present study. ACE inactivates bradykinin, a potent vasodilator and catalyses the conversion of angiotensin I to angiotensin II levels which stimulates sodium and water reabsorption in the kidneys. Furthermore, angiotensin II increases peripheral resistance and stimulates aldosterone secretion which also has sodium and water retaining effects cumulatively increasing blood volume and consequently blood pressure [3, 18]. The high penetrance of the DD genotype in African populations could explain why ACE inhibitors are ineffective first-line treatments among Africans resulting in more resistant forms of hypertension compared to other

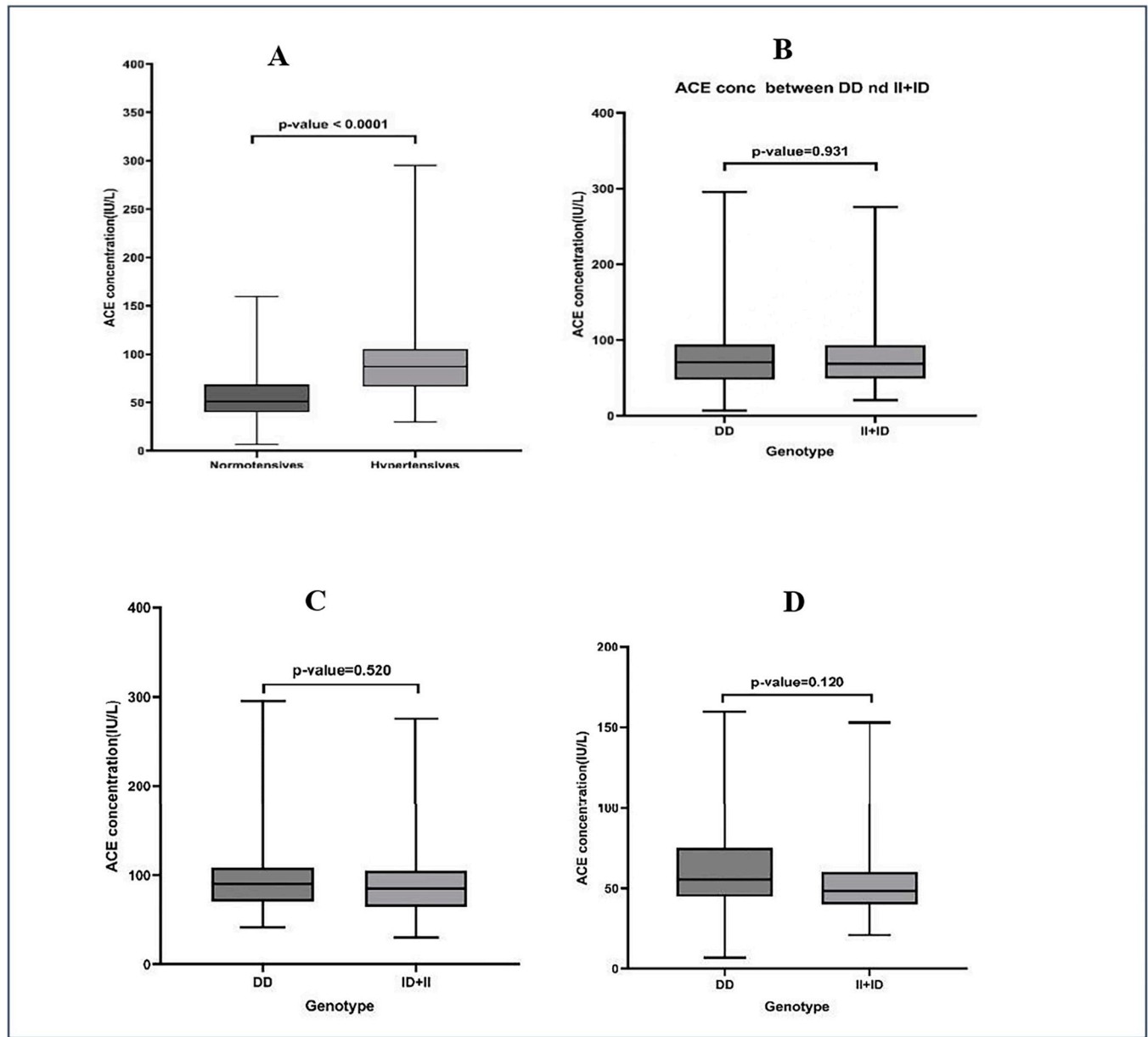

**Fig 2.** Comparison of ACE levels between hypertensives vs normotensives (**A**), DD vs II+ID among study participants (**B**), DD vs II+ID among normotensives (**C**) and DD vs II + ID among hypertensives (**D**).

races [25, 32]. By contrast, several studies among Gabonese [33], Tunisians [16, 17], Spanish [18] and Chinese [7] did not report a significant association with some studies even reporting an inverse association [20]. The contradictory results concerning the involvement of ACE I/D polymorphisms in HTN may be due to ethnic variations, heterogeneous populations, geographical variations, sampling biases, and possibly other ecological factors [12]. In addition, some environmental factors, such as nutrition and physical activity, are associated with changes in the epigenetic status [34].

In the present study, while plasma ACE levels were significantly elevated among hypertensives compared to controls, ACE levels did not significantly differ between the DD genotype

and II+ID genotypes among hypertensives and normotensive subjects as proposed by some studies that suggests that 47% of phenotypic plasma ACE diversity is attributable to the ACE I/D polymorphism [35]. [36] also did not find any association between ACE levels and hypertension in men and women. This could be due to the DD genotype that significantly influences tissue (kidneys and lungs) ACE levels more strongly than the plasma ACE levels estimated in this study [30, 37, 38]. ACE levels in the kidneys and heart are generally at moderate levels and an increase in levels(independent of serum ACE) significantly increase the rate of angiotensin II formation that can cause hypertension [39].

## 5. Conclusion

The current study has shown that ACE I/D polymorphism is associated with the risk of developing hypertension. The ACE DD genotype and D significantly increases the risk of developing hypertension in the Ghanaian population. As a result, ACE I/D polymorphism may be used as a biomarker for early diagnosis, detection and prevention of hypertension complications.

## 6. Limitations

This study did not include all other polymorphisms within the ACE gene which could be in linkage disequilibrium with the I/D polymorphism. Additionally, only serum ACE was estimated, intracellular ACE was not estimated and could also be involved in the RAAS pathway.

## Supporting information

**S1 Checklist. Strobe checklist.**
(DOC)

**S1 Dataset. Minimal data set.**
(XLSX)

**S1 Raw images. Gel raw images.**
(PDF)

**S1 File. WHO step questionnaire.**
(DOC)

## Acknowledgments

The authors would like to acknowledge the Department of Molecular Medicine, Kwame Nkrumah University for Science and Technology for providing the laboratory equipment and facilities for conducting molecular analysis.

## Author Contributions

**Conceptualization:** Precious Bonney.

**Data curation:** Yussif Adams, Jemimah Duodu.

**Formal analysis:** Precious Bonney, Lawrence Quaye.

**Funding acquisition:** Precious Bonney, Peter Paul Dapare.

**Investigation:** Precious Bonney, Grace Bourawono, Mercy Akaluti, Jemimah Duodu.

**Methodology:** Precious Bonney, Peter Paul Dapare, Grace Bourawono.

**Project administration:** Christian Obirikorang.

**Resources:** Grace Bourawono, Mercy Akaluti.

**Supervision:** Christian Obirikorang.

**Visualization:** Lawrence Quaye.

**Writing – original draft:** Precious Bonney, Yussif Adams.

**Writing – review & editing:** Precious Bonney, Lawrence Quaye, Peter Paul Dapare, Jemimah Duodu.

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
