## [Decision Letter · Decision Letter 0]

1 Aug 2024

PONE-D-24-24207Association between Angiotensin-(ACE) insertion/deletion polymorphism and hypertension in a Ghanaian populationPLOS ONE

Dear Dr. Bonney,

Thank you for submitting your manuscript to PLOS ONE. After careful consideration, we feel that it has merit but does not fully meet PLOS ONE’s publication criteria as it currently stands. Therefore, we invite you to submit a revised version of the manuscript that addresses the points raised during the review process.

We look forward to receiving your revised manuscript.

Kind regards,

Sepiso Kenias Masenga, PhD

Academic Editor

PLOS ONE

5. We notice that your supplementary figures are uploaded with the file type 'Figure'. Please amend the file type to 'Supporting Information'. Please ensure that each Supporting Information file has a legend listed in the manuscript after the references list.

6. Please upload a copy of Fig 1-Gel and Fig 1- ACE , to which you refer in your text on page 7 and 12. If the figure is no longer to be included as part of the submission please remove all reference to it within the text.

Additional Editor Comments:

1. Please follow the STROBE checklist and add it as supplementary file

2. We note that your Data Availability Statement is currently as follows: All relevant data are within the manuscript and its supporting information files.

3. When completing the data availability statement of the submission form, you indicated that Yes - all data are fully available without restriction. We strongly recommend all authors decide on a data sharing plan before acceptance, as the process can be lengthy and hold up publication timelines. Please note that, though access restrictions are acceptable now, your entire data will need to be made freely accessible if your manuscript is accepted for publication. This policy applies to all data except where public deposition would breach compliance with the protocol approved by your research ethics board. If you are unable to adhere to our open data policy, please kindly revise your statement to explain your reasoning and we will seek the editor's input on an exemption. Please be assured that, once you have provided your new statement, the assessment of your exemption will not hold up the peer review process.

Reviewers' comments:

Reviewer's Responses to Questions

**Comments to the Author**

1. Is the manuscript technically sound, and do the data support the conclusions?

Reviewer #1: Yes

Reviewer #2: Partly

2. Has the statistical analysis been performed appropriately and rigorously? 

Reviewer #1: Yes

Reviewer #2: Yes

3. Have the authors made all data underlying the findings in their manuscript fully available?

Reviewer #1: Yes

Reviewer #2: Yes

4. Is the manuscript presented in an intelligible fashion and written in standard English?

Reviewer #1: Yes

Reviewer #2: Yes

5. Review Comments to the Author

Reviewer #1: Article Title

Association between Angiotensin-(ACE) insertion/deletion polymorphism and hypertension in a Ghanaian population

Study Summary

This case-control study conducted by Bonney et al. at Tamale Central Hospital investigates the relationship between the angiotensin-converting enzyme (ACE) insertion/deletion (I/D) polymorphism and essential hypertension in the Ghanaian population. The study sample size was 144 participants, comprising 72 hypertensive patients and 72 normotensive individuals. The finding of the study was that the DD genotype was significantly associated with an increased risk of hypertension. The finding suggests the potential of ACE I/D polymorphism as a biomarker for hypertension.

Major Strengths of the study

The study effectively addresses a significant gap in the literature by focusing on the Ghanaian population, which is underrepresented in genetic studies on hypertension. The case-control design and the utilization of logistic regression to adjust for confounders enhance the validity of the study findings.

Minor Comments

1. The authors should consider adding a section on the limitations of the study, including potential biases, to increase transparency and help readers critically evaluate the findings within the appropriate context.

Reviewer #2: Manuscript/paper title: Association between Angiotensin-(ACE) insertion/deletion polymorphism and hypertension in a Ghanaian population.

Summary statement of the article:

The article explores the relationship between ACE I/D polymorphism and essential hypertension among patients in Ghana. This case-control study involved 144 participants, including 72 hypertensive patients and 72 normotensive individuals. The findings revealed that the DD genotype significantly increased the risk of hypertension, with an adjusted odds ratio of 8.52. The study concluded that the presence of the DD genotype is strongly associated with an increased risk of hypertension in the Ghanaian population. The research highlights the importance of genetic factors in the pathophysiology of hypertension and suggests that ACE I/D polymorphism could serve as a biomarker for early diagnosis and prevention of hypertension complications.

Specific areas of improvement

Major comments:

1. I encourage the authors to rewrite the abstract, i must say it is well written and concise but adding some more flesh would make the article more credible.

2.Citations: Authors should ensure that all statements are cited, as this rules out the practice of plagiarism in the article. Starting with the first statement under the “Introduction section.”

3. Statistical Analysis: Authors should provide more detailed explanations of the statistical methods used, including any software or tools employed for the analysis. They have mentioned some statistical tests that were employed but not the specific tools/software, e.g. SPSS/STATA version 1. Etc...

4. Under the data analysis section, authors have emphasized that they considered data to be statistically significant at P-value < 0.05. However, this has not been shown in Table “6: Logistic regression analysis of the ACE polymorphism.” Please show the P-Values in Table 6, for clarity and proof that indeed the results truly represent what has been researched.

5. Figures and Tables: Ensure that all figures and tables are clearly labelled and referenced in the text. Consider uploading clearer figures to enhance the presentation of data.

• “Table 1: Primer sequence used in PCR amplification of ACE I/D polymorphism.” If authors could try to make clearer narrations, it would be hard to understand.

• Figure 1, insert it closer to where the description is. Do this for all tables and figures.

• Authors should consider the input of narrations to all tables attached to this article that are articulate and easy to follow.

• All the figures that have been uploaded are not clear including their narrations. It makes readers think that they were downloaded from the internet and pasted. Authors, please work on that too.

6. Under 2.3 Data collection procedure, the authors must ensure to include the (WHO Step questionnaire for chronic diseases) and the WHO staging tool that has been referred to have been used. I advise that these be included as part of the supplementary files or provide a link or cite, as this will help in the clarity of the article to the readers.

7. Under 2.3 Data collection procedure, the statement - “BMI was classified into five categories according to the WHO (2008) guidelines: underweight (<18.5 kg/m-2), normal (18.5-24.9 kg/m-2), overweight (25.0 - 29.9 kg/m-2) and obese (≥30.0 kg/m-2).” The authors have mentioned that they used 5 categories of classification, but only 4 have been mentioned, please clarify this.

8. Discussion Depth: Expand the discussion to include more comparisons with similar studies in other populations, highlighting the significance of the findings in the context of existing literature.

Minor Corrections:

1. Authors should ensure any terminologies are defined at first use. E.g. K3EDTA has not been defined. Please look into all terms that might have been left out.

2. Check for any grammatical errors or awkward phrasing in the summary. Clarity and readability are essential.

3. What were the limitations and strengths of this research? Adding these sections will improve the clarity of the article including its reproducibility.

4. Consider page alignment and justification in the article, especially for the results tables. Authors can consider other tables being in a landscape mode.

6. PLOS authors have the option to publish the peer review history of their article (what does this mean?). If published, this will include your full peer review and any attached files.

Reviewer #1: **Yes: **Lweendo Muchaili

Reviewer #2: **Yes: **Bislom Chikwanka Mweene

---

## [Author Response · Author response to Decision Letter 0]

18 Sep 2024

As per the corrections and suggestions requested on the aforementioned manuscript, the following changes have been made;

1. Journal requirements: All formatting requirements especially file naming errors have been corrected to meet PLOS ONE style requirements. Additionally, I can confirm that all relevant raw data including uncropped blot/gel results as well as the minimal data set have been uploaded as supplementary information files in this submission. The ethical considerations section have also been modified to include the IRB that approved the study and the information of the written consent. The supplementary figures were wrongly labeled, and the revised submission has been corrected to reflect the appropriate names of figures and supplementary files. A copy of the Fig-1 Gel and Fig 1 ACE referred to on page 7 and 12 respectively has been uploaded. The reference list has been curated to ensure currency and accuracy.

2. Additional Editor comments: The strobe cheklist per your request has been followed and uploaded as a supplementary file. The entire raw data has been made available as a supplementary file.

3. Reviewer 1 Comments: The limitations of the study have been included in the revised manuscript.

4. Reviewer 2 Comments: The abstract has been modified to increase the credibility of the article. However, the author appreciates the word limit according to PLOS ONE formatting style and therefore worked within those limits. Citations and references have also been checked throughout the document to rule out plagiarism. The statistical analysis section has been edited to include a more detailed description of the tools and statistical methods used. The p-values generated per the logistic regression model used to generate the results table 6 have been added to the table. The labels for figures and tables have been changed where necessary to provide more clarity on the corresponding tables and figures. The WHO step questionnaire has been uploaded as a supplementary file. The errors with regards to the number of categories of classification of weight categories used in the study is four and not five and has been modified to reflect same.The discussion section has been expanded to include more comparisons with similar studies in other populations.

5. Reviewer 2 minor corrections: Terminologies and acronyms which were previously not defined at first use have been corrected. Grammatical errors have been checked to improve readability. Page alignment and justification in the article have been corrected where necessary.

---

## [Editor Report · Decision Letter 1]

24 Sep 2024

Association between Angiotensin-converting enzyme (ACE) insertion/deletion polymorphism and hypertension in a Ghanaian population

PONE-D-24-24207R1

Dear Dr. Bonney,

We’re pleased to inform you that your manuscript has been judged scientifically suitable for publication and will be formally accepted for publication once it meets all outstanding technical requirements.

Kind regards,

Sepiso K. Masenga, PhD

Academic Editor

PLOS ONE
---

## [Editor Report · Acceptance letter]

1 Oct 2024

PONE-D-24-24207R1 

PLOS ONE

Dear Dr. Bonney, 

I'm pleased to inform you that your manuscript has been deemed suitable for publication in PLOS ONE. Congratulations! Your manuscript is now being handed over to our production team.

Kind regards, 

on behalf of

Prof. Sepiso K. Masenga 

Academic Editor

PLOS ONE